# Inducing Angiogenesis, a Key Step in Cancer Vascularization, and Treatment Approaches

**DOI:** 10.3390/cancers12051172

**Published:** 2020-05-06

**Authors:** Harman Saman, Syed Shadab Raza, Shahab Uddin, Kakil Rasul

**Affiliations:** 1Barts Cancer Institute, Queen Mary University of London, London E1 4NS, UK; 2Department of Medicine, Hazm Maubrairek Hospital, Ar-Rayyan PO Box 305, Qatar; 3Department of Stem Cell Biology and Regenerative Medicine, ERA University, Lucknow 226003, India; Drshadab@erauniversit.In; 4Translational Research Institute, Academic Health System, Hamad Medical Corporation, Doha 3050, Qatar; Skhan34@hamad.qa; 5National Cancer Care and Research, Hamad Medical Corporation, Doha 3050, Qatar; Krasul@hamad.qa

**Keywords:** angiogenesis, cancer, VEGF, anticancer

## Abstract

Angiogenesis is a term that describes the formation of new blood and lymphatic vessels from a pre-existing vasculature. This allows tumour cells to acquire sustenance in the form of nutrients and oxygen and the ability to evacuate metabolic waste. As one of the hallmarks of cancer, angiogenesis has been studied extensively in animal and human models to enable better understanding of cancer biology and the development of new anti-cancer treatments. Angiogenesis plays a crucial role in the process of tumour genesis, because solid tumour need a blood supply if they are to grow beyond a few millimeters in size. On the other hand, there is growing evidence that some solid tumour exploit existing normal blood supply and do not require a new vessel formation to grow and to undergo metastasis. This review of the literature will present the current understanding of this intricate process and the latest advances in the use of angiogenesis-targeting therapies in the fight against cancer.

## 1. Introduction

Under physiological conditions, angiogenesis is a highly regulated process. It plays crucial roles in embryogenesis, wound healing and the menstrual cycle [1]. Angiogenesis is also seen in non-malignant pathologies such as diabetic retinopathy, ischaemic diseases and autoimmune conditions such as connective tissue diseases and psoriasis [1].

In addition to providing nutrients and oxygen to the tumour and the removal of metabolic waste, new vessel formation also enables cancer cells to metastasize and proliferate to distant sites through entry into the newly formed blood and lymphatic system and subsequent extravasation [2]. A lack of adequate blood supply, on the other hand, could halt tumour growth, and might even lead to tumour shrinkage and sometimes cancer cell death [3]. Previous studies demonstrated that, in the absence of angiogenesis, tumours could grow to a maximum of 1–2 mm^3^ in diameter before they stopped growing and died, whilst some tumour cells could grow beyond 2 mm^3^ in size in angiogenesis-rich cell culture. The continued growth of cancer cells in angiogenesis-rich cell culture is explained by reproducing physiological properties in a three-dimensional cell culture model that provides controlled fluid perfusion that permits the regulation of oxygen intake, promoting a circulatory environment that is controlled by computer hardware [4].

## 2. Angiogenesis in Normal Tissue

The structure of the blood vessels depends on their size; small blood vessels are comprised of endothelial cells (EC), whereas in medium and large blood vessels, ECs are surrounded by pericytes (mural cells) [5]. In normal tissue, the process of neovascularization is tightly controlled. The process includes stepwise stages (Figure 1).

After this strictly controlled vessel formation, the normal vasculature becomes largely quiescent [5]. Angiogenesis is controlled by several growth factor stimulators and inhibitors. Angiogenic (stimulatory) growth factors include Fibroblast Growth Factor, Granulocyte Colony-Stimulating Factor, Interleukin-8, Transforming Growth Factors alpha and beta and Vascular Endothelial Growth Factor. Angiogenic inhibitors include Angiostatin, Interferons (alpha, beta and gamma), Endostatin, Interleukin-12 and retinoids [5]. Inhibitory factors are present within the extra-cellular matrix (ECM). At a molecular level, angiogenesis is normally controlled by a family of small none-coding RNA molecules that are collectively called angiomiRs. AngiomiRs are comprised of pro-angiogenic miRs and anti-angiogenic miRs (Table 1) [6]. A well-studied angiomiRs is miR-200b, which belongs to the miR-200 family [7]. miR-200b has antiangiogenic effects. Its expression is transiently turned down when new vessel formation is required, for example during wound-healing. Once the physiological demand subsides, miR-200b is expressed again to stop angiogenesis as a measure of tight control on new vessel formation. The downregulation of miR-200b in response to tissue hypoxia triggers epithelial to mesenchymal transition and modulates endothelial cell migration which result in new vessel formation [8]. There is evidence that the dysregulation of iR-200b contributes to oncogenesis and metastasis in some cancers, such as breast cancer [9].

## 3. Angiogenesis in Cancer, a Literature Review

In cancer, a switch to angiogenesis seems to be an imbalance between stimulatory and inhibitory factors that leads to a pro-angiogenic state [18]. This results from a state of a relatively poorly blood-supplied hyperplasia converted to an uncontrollable new vessel formation that ultimately causes malignant tumour progression. Researchers have investigated the molecular basis of pro- and inhibitory pathways with the view of better understanding oncogenesis and the development of anti-cancer treatment. The flip side of angiogenesis is poor tumour blood supply. Poor tumour blood supply is one of the postulated mechanisms of resistance to chemotherapy, due to the failure of an adequate delivery of cytotoxic drugs to the tumour site [19]. For example, for decades the five year overall survival of pancreatic cancer has not exceeded 5%, despite extensive research [20]. One explanation for this is that pancreatic cancer tissue is surrounded by dens stromal tissue that hinders the delivery of anticancer therapies. In contrast to antiangiogenetic treatment, vascular promotion therapy is investigated to promote tumour blood supply to facilitate the better delivery of cytotoxic drugs to the target tissue [20].

## 4. Pro- and Anti-Angiogenic Factors

Judah Folkman coined the phrase tumour angiogenesis and studied this process extensively [18]. He led the discovery of the first angiogenic factors. These factors trigger neovascularization through inducing angiogenesis switch [21]. As seen in Figure 2, tumour overgrowth is believed to be halted through maintaining an equilibrium between pro- and anti-angiogenesis factors, leading to a state of tumour dormancy [18,21].

Disturbance to this equilibrium results in increased angiogenesis, and thus uncontrollable tumour overgrowth [22]. Several angiogenic factors have been described. Of those, vascular endothelial growth factor A (VEGF-A) is a major regulator of angiogenesis both under normal conditions and in disease state [11]. VEGF-A belongs to family of gene factors that also encompasses VEGF-B, VEGF-C, VEGF-D, VEGF-E and placenta growth factor (PlGF). These growth factors have different levels of specificity and different affinities to tyrosine kinase receptors (VEGFR) 1,-2 and -3 [22]. The binding of VEGF-A to VEGFR 2 (predominantly found on EC of blood vessels) leads to angiogenesis, whereas VEGF-C and D preferentially bind to VEGFR-3, expressed predominantly on lymphatic EC, resulting in the proliferation of lymphatic vessels [23]. In cancer, the role of VEGF exceeds angiogenesis through a complex autocrine and paracrine signaling pathway; VEGF plays an important role in promoting the cancer stem cells’ functionality and the initiation of tumour [24]. The upregulation of VEGF initiates tumourigenesis by contributing to the activation of epithelial–mesenchymal transition (EMT) [25]. EMT represents a key event in the process of new vessel formation. [26]. This because EMT leads to a loss of cell polarity and dramatic cytoskeletal changes, which lead to increased cell motility and loss of cells to cell adhesion by the loss of E-cadherin and ZO-1. The last two markers are associated with epithelial cells. EMT also results in the production of several proteolytic enzymes, including matrix metalloproteases and serine proteases that degrade the extracellular matrix (ECM). Several pathways involved in EMT support endothelial cell (EC) survival and proliferation [26]. These pathways invlove complex interactions between the cell membrane, ECM and intracellular regulatory signalling pathways. The resulting phenotypical changes caused by EMT promote cancer cell invasion of basement membrane, and eventually cancer cell metastasis [27].

Moreover, in hypoxic tumour, tumour-associated macrophages (TAMs), known for their protumour functions, secrete VEGF. VEGF interacts with key immune cells in the tumour micro-environment (TME), namely CD4+ forkhead box protein P3 (FOXP3) + regulatory T cells, which a strong suppressor of anticancer immunity. VEGF is able to attract these regulatory T cells to the TME using the chemoattractant neuropilin 1 (NRP1). Animal studies showed that removing NRP1 was associated with increased infiltration of TME with antitumoural CD8 T cells, with a reduction in tumour growth. Fibroblasts, that are present in abundance in TME and known to support tumour growth, also secrete VEGF. VEGF, in turn, stimulates fibronectin fibril assembly; the latter has a potent protumour effect within the TME [28].

Other angiogenesis promotors include platelet-derived growth factor (PDGF)-B and C and fibroblast growth factor (FGF)-1 and -2 [29]. Both groups of factors exert their effect, EC proliferation and migration, once they bind to their respective receptors on blood vessels’ EC.

As seen in Figure 3, Tie1 and Tie2 are two signaling pathways that encompass the interaction between angiopoietins, tyrosine kinases (TK), VEGF and their receptors [29].

Other important regulators of angiogenesis are angiopoietins. Angiopoietins interact with Tie-2 TK receptor found on EC. Through cooperating with other angiogenesis factors, angiopoietins modulate the activity of the EC [29,30]. Angiopoietin-1 (Ang-1) and angiopoietin-2 (Ang-2) can form dimers, trimers and tetramers. Angiopoietin-1 can form higher-order multimers through its super clustering domain. It is believed that not all these structures bind with the TK receptor; the only activators of these receptors are at the tetramer level or higher [30].

Other promotors of angiogenesis include a wide range of polypeptides, metabolites and hormones that contribute to new blood vessels’ formation in both physiological and disease state [30]. On the other hand, there is a wide range of antiangiogenetic factors that oppose the function of the promotors. Constituents and proteolytic fragments of the extra-cellular matrix (ECM) and the basement membrane represent potent angiogenesis inhibitors [31]. A well-studied angiogenesis inhibitor is thrombospondin-1 (TSP1), which is a large glycoprotein present in ECM [31]. Another matrix-derived angiogenesis inhibitor is a proteolytic product of collagen XVIII called endostatin [32]. Interferon-alpha and -beta and angiostatin, a cleavage product of plasmin, are other examples of angiogenesis inhibitors [33].

The activities of both angiogenesis promoters and inhibitors are regulated through a complex interaction of different pathways. The proangiogenic imbalance often occurs at the gene level due to the activation of oncogenes or inactivation of tumour suppressor genes, all the way to cell environmental factors such as hypoxia, hypoglycaemia, cellular nutrient deficiency and metabolic acidosis [34]. As part of multistage tumourigenesis, angiogenic switch arises from an imbalance between pro- and inhibitors of angiogenesis activity and level; this imbalance is driven from the tumour cells and the inflammatory cells that infiltrate the tumour [35]. The next section will focus on the mechanisms behind angiogenic switches in cancer and the different pathways involved.

## 5. Angiogenic Switch

In a seminal paper, Judah Folkman, Doug Hanahan and colleagues presented angiogenic switch in a transgenic Rip1Tag2 mouse module of pancreatic beta-cell carcinogenesis, during the progression from hyperplasia to heavily vascularised cancer [36]. Rip1Tag2 mice express the Simian Virus 40 large T antigen oncoprotein under the control of the rat insulin promoter. This leads to the overexpression of oncogene in pancreatic beta-cells of the islets of Langerhans, resulting in the development of beta-cell tumour. The study of Rip1Tag2 mice showed the phases of tumour genesis, from normal cells to hyperplasia, and adenoma to invasive carcinoma. VEGF-A was shown to be the main driver of EC proliferation, migration and tube formation, all essential components of angiogenesis [36]. Mice that overexpressed human VEGF-A165 in pancreatic beta-cells had angiogenesis at an early stage of tumourigenesis [37]. In contrast, inhibiting VEGF-A resulted in suppressing angiogenic switch and tumour growth [38,39]. Different techniques were used to inhibit VEGF-A, such as chemical inhibitors of VEGFR signaling or genetically depleting VEGF-A in beta-cells [39,40]. Figure 4 is a schematic representation of angiogenic switch in transgenic mice; it shows progression from dormant hyperplasia to growing hyper-vascularized tumours as the result of angiogenesis.

Another important element of angiogenic switch is stromal cells of tumour microenvironment [41]. Through chemotaxis, cancer cells recruit innate immune cells. The immune cells contribute to angiogenesis via secreting pro-angiogenic factors. Using paracrine stimulation, tumour-associated macrophages (TAM) partake in the modulation of angiogenesis and tumour progression [42]. The cytokine/chemokine component of the tumour microenvironment determines the function of TAM. This function is either the M1 state of macrophages, which is an anticancer, or the M2 state, which suppresses immunity and promotes tumourigenesis via secreting pro-angiogenic cytokines and VEGF-A [43].

Endothelial progenitor cells (EPC), are also believed to play a role in angiogenesis [44]. Tumour-secreted factors recruit EPC from bone marrow to the tumour site to contribute to angiogenesis [45]. However, the exact role of EPC in angiogenesis remains to be fully understood [44,45]. Studies of mouse models of breast cancer have shown that myeloid progenitors differentiated to EC, leading to neovascularization [46]. This is further evidence of the role of the immune cells in promoting angiogenesis.

Importantly, not all tumours rely on new blood vessel formation to survive and grow [47], and therefore the angiogenic switch might never occur. Some tumours exploit the existing blood supply through a process named vessel co-option to support their growth and to enable metastasis. Vessel co-option has been observed in a number of tumours such as non-small cell lung cancer (NSCLC), glioblastoma and hepatocellular carcinoma [48,49]. Cancer cells seem to grow along existing vessels and/or invade the connective tissue that is present between the vessels, allowing the cancer cells to incorporate to the existing normal vasculature to begin hijacking the blood supply [50,51]. There is evidence that vessel co-option promotes cancer cell motility and metastasis and tumour dormancy [52]. Moreover, some tumours such as NSCLC, use both angiogenesis and vessel co-option simultaneously or sequentially (in no particular order) to acquire blood supply and venous and lymphatic drainage [53]. Moreover, there is growing evidence that increased vascularity, often measured through microvascular density, caused by vessel co-option, is associated with higher tumour grade and higher risk of metastasis [54]. Interestingly, bone marrow appears to be an important site for vessel co-option in both primary and secondary bone malignancies which, in turn, might explain the development of tumour dormancy in bones and the higher rate of chemoresistance [55,56].

In addition there is also evidence from preclinical studies that show that some tumors, such NSCLC and gliomas, never undergo angiogenic switch and rely only on vessel co-option [53,57]. In contrast, some tumors, for example hepatocellular carcinoma and liver metastases of the gastrointestinal tract, switch from using vessel co-option at early stages of tumourigenesis to angiogenesis at a later stage during tumour progression [58,59]. This progression from vessel co-option to angiogenesis is not an obligatory requirement of tumour progression and metastasis [60]. Moreover, preclinical and clinical studies showed that there are, at times, but not always, differences that exist between primary and secondary versions of of the same tumour in terms of their access to blood supply [61,62]. For example, when cells from angiogenic primary human breast tumors spread to the lung tissue, they switch to vessel co-option as a mode of accessing blood supply [62], which also functions as a resistant mechanism against antiangiogenic therapy [54].

## 6. Tumour Vasculature Modulation as a Therapeutic Option

### Vascular Promotion Therapy

This approach is presumed to work though improving the delivery of cytotoxic agent(s) to the tumour (Figure 5).

An example of this is the use of Cilengitide and Verapamil in conjunction with Gemcitabine or Cisplatin to treat pancreatic ductal adenocarcinoma [63]. In high doses, Cilengitide, a selective inhibitor of integrins, leads to inhibition of the FAK/SRC/AKT pathway, causing apoptosis in EC. This drug was originally developed as an antiangiogenic agent. However, in clinical trials, it showed no efficacy in the treatment of glioblastoma. A low dosing of Cilengitide was, however, observed to be associated with the promotion of tumour angiogenesis [64]. Verapamil, a calcium channel blocker, causes vasodilatation, hence the increased blood flow to tumour. Cilengitide and Verapamil, in addition to Gemcitabine administered in a xenograft tumour model, through various schedules, mimicking human dosing regimens, was studied in trials by Wong et al. [63].

The impact of the therapy on the tumour blood flow was assessed by flow cytometry, imaging techniques and the concentration of the drugs in vital organs, tumour, and blood levels. The studies showed increased functional (less leaky) vessel formation, leading to an improved tumour blood supply to both highly and poorly vascularized tumors. This effect translated into tumour regression and improved survival in vivo models. The authors showed that vascular promotion increased the cell uptake of Gemcitabine with reduced side effects. The authors also argued that the promotion of vascularization improved the efficacy of Cisplatin due to better tumour blood perfusion, which improved cytotoxic delivery, leading to tumour regression in mice model. Future studies should deal with the impact of this approach in different tumour sites and their secondaries, address the wide variations in tumour behavior caused by intratumor heterogeneity and focus on the potential complications of promoting neovascularization, such as the risk of significant/life threatening bleeding, and its safety in vascular diseases.

## 7. Immune Modulation

As mentioned, the infiltration of tumour microenvironment with immune cells, importantly TAM, is associated with pro-angiogenic factor secretions by these cells. Several experiments studied the inhibition of TAM function or their complete removal from the tumour microenvironment. A study showed that treating K14-HPV/E2 mice with Zoledronic acid (ZA), a bisphosphonate used for skeletal metastasis with anti-inflammatory and anti-osteoclast properties, resulted in the suppressed mobilization of VEGF-A and, consequently, the inhibition of angiogenesis and tumourigenesis [65]. Other studies showed that treatment with ZA in advanced solid tumors was associated with a reduction in VEGF-A plasma levels [66]. The inhibition of neutrophils and macrophages to reverse angiogenic switch has been tested in preclinical trials but not applied in clinical settings [67]. Other immune modulated strategies that have been studied include: inhibition of Cyclooxygenase-2 (COX-2) expression by COX-2 inhibitors in pancreatic and cervical cancer [68] and Lenalidomide (an immunomodulatory drug) in advanced renal cancer [69], with benefits in phase two trials but no additional advantage in combination with standard cytotoxic protocols.

## 8. Anti-Angiogenic Therapy

The Food and Drug Administration (FDA) approved biological therapies in the form of tyrosine kinase inhibitors (TKIs), monoclonal antibodies and fusion peptides in non-small cell lung cancer, metastatic colorectal cancer, medullary thyroid cancer and renal cell cancer [70]. More specifically, targeting VEGF has become an important approach to stop tumour growth (Figure 6), and part of the treatment protocol of several tumour primaries, notably colon, non-small cell lung and renal cell cancers [71]. Several studies showed that arteriol formation and tortuosity, as well as venous dilation, are increased through VEGF expression [72]. Cell culture injected with adenovirus expressing VEGF undergo the induction of mother vessels (MV) and stabilized MV from normal capillaries and venules. In contrast, the inhibition of VEGF is shown to cause veins and arterioles to have fewer cleavage planes. For example, Aflibercept, a decoy receptor that binds VEGF-A, induces the rapid collapse of mother vessels (MV) to glomeruloid microvascular proliferations (GMP). VEGF inhibition, by anti-VEGF/VEGF receptor, is shown to restore vasculature within hours to normal microvessels by way of GMP [73]. GMP is believed to act as an intermediary step in MV reversion to normal microvessels after VEGF blockade [74].

Monoclonal antibodies such as Bevacizumab, which blocks the VEGF receptor, or small molecules such as Lapatinib, which inhibits TK downstream of VEGF, are examples of anti-VEGF treatment. Phase 1 trial of Bevacizumab showed that the drug was well tolerated and had good pharmacokinetic properties [75]. A phase 3 clinical trial of Bevacizumab in metastatic colorectal cancer (mCRC) showed a modest impact of 4 to 5 month improvement in overall survival (OS) in metastatic colon cancer [76]. In transgenic mouse models of non-squamous non-small cell lung cancer (nsNSCLC), Bevacizumab was shown to reduce the risk of brain metastasis, and therefore improve survival. This might translate into improved survival due to a reduction in the rate of brain metastases in patients with stage III nsNSCLC [77]. Despite prolonging the PFS of metastatic breast cancer, the FDA removed Bevacizumab from standard treatment protocol due to safety concerns [78].

Combining Bevacizumab with chemotherapy, in the first and second line settings of mCRC, improved OS [79]. The AVF2107g study showed an improvement in median survival from 15.6 to 20.3 months when combining Bevacizumab to irinotecan, bolus fluorouracil, and leucovorin, compared to placebo [76] in treatment-naïve mCRC patients. PFS, but not OS, was shown to improve in a randomized controlled trial of mCRC combining Bevacizumab with oxaliplatin-based chemotherapy as first-line treatment [80]. Another randomized controlled trial showed that adding Bevacizumab to fluorouracil and leucovorin improved PFS in patients with mCRC for whom first-line irinotecan was judged inappropriate due to their poor functional status [81]. The direct VEGFR2 antagonist, Ramucirumab, was approved in the treatment of advanced hepatocellular carcinoma (HCC) with high alpha-feto protein after progression to sorafenib [82]. Through binding to VEGF-B and placental growth factor, Ziv-aflibercept, a representative agent of the third type of angiogenesis inhibitor, composed of the extracellular domain of both VEGFR-1 and VEGFR-2 fused to the Fc region of IgG1, inhibits the pro-angiogenic effects of the VEGF/VEGFR signaling pathway [83]. Ziv-aflibercept, in combination with 5-fluorouracil, leucovorin and irinotecan (FOLFIRI) for mCRC, in patients resistant to or progressing after treatment with oxaliplatin, showed statistically significant improvements in PFS and OS [84].

Given the results of animal trials, this modest benefit of anti–VEGF-A/VEGFR therapy against human cancers has been relatively disappointing. One explanation for this modest effectiveness is that most cancer patients are elderly, frail and cannot tolerate high doses, in contrast to relatively healthy tumour-bearing mice that can be given higher doses [85]. Another possible reason is that tumour hypoxemia resulting from anti–VEGF-A/VEGFR therapy lead to the over-expression of matrix components that bind and sequester VEGF-A, rendering anti-VEGF drugs ineffective [86]. Hypoxia also might stimulate cancer cells to secrete other pro-angiogenic factors such as FGF, PDGF-B, PDGF-C, HGF, EGF, IL-8, IL-6, Ang-2, SDF1a, PDGF-C, CXCL6 and others, as well as their receptors [85,86]. Mobilisation from bone marrow to the tumour site of vascular progenitor cells and proangiogenic myelocytes are other mechanisms that might be responsible for the limited effectiveness of anti–VEGF-A/VEGFR therapy [87]. Another hindrance to anti-angiogenesis therapy is that the blood supply to the tumour is reduced, and this would lead to the impairment of the delivery of chemotherapy agents to the tumour, hence reducing their cytotoxic effects. Antiangiogenic treatment creates a hypoxic tumour microenvironment, which results in the tumour cells becoming more “aggressive” and promotes “escaping” of the tumour cells from the hypoxic environment to distant, normo-oxic, sites, i.e., metastasis [88]. Other mechanisms of therapy resistance involve the recruitment of pro-growth cells and molecules to the TME by the cancer cells as the result of tumour hypoxia, such as tumour-associated macrophages [89], tumour-associated fibroblasts (TAFs) [90], Tie2^+^ monocytes [91], myeloid cells [92], pro-angiogenic bone-marrow-derived cells including CD11b^+^ Gr1^+^ and the overexpression of alternative angiogenic signaling molecules [93], including a fibroblast growth factor-2 [94], interleukin-8 (IL-8) [95], IL-17 [96], and angiopoietin 2 [97].

Vessels’ co-option as a mechanism to attain blood supply by cancer cells is another resistant mechanism to anti-angiogenic treatment. Preclinical models demonstrated a switch from angiogenesis to vessel co-option during anti-angiogenic treatment [98,99]. The escaping anti-angiogenic agents’ effect using vessel co-option is seen across a range of cancer types. For example, the modest response of glioma to bevacizumab is shown, in preclinical studies and clinical case reports, to be due to vessel co-option [100]. This could be intrinsic resistance or acquired during treatment with bevacizumab due to the switch from angiogenesis to the vessel co-option [101,102]. This switch from angiogenesis to the vessel co-option is also observed during the treatment of breast cancer with anti-angiogenic therapy. Pulmonary metastasis from breast cancer is shown to use the lung parenchymal blood supply for their survival and growth, which explains their resistance to anti-angiogenic therapy [61]. In addition, preclinical trials showed that after an initial response of xenograft model of hepatocellular carcinoma to sorafenib (a multi-kinase inhibitor with antiangiogenic properties), the tumour progressed within a month due to the large-scale co-option of sinusoidal and portal tract vessels [54]. Moreover, several studies showed that resistance to anti-angiogenic therapy in metastatic colorectal carcinoma (CRC) to the liver is likely secondary to the CRC cells co-option of pre-existing liver vessels; this can occur in the context of both intrinsic and acquired resistance [60].

## 9. Novel and Future Approaches to Modify Angiogenesis as Anti-Cancer Option

Targeting angiogenesis has shown limited effectiveness to date, but affirms Folkman’s postulations. This limited success is likely caused by the heterogeneity of blood vessels, as some vessels are susceptible, whilst others are resistant, to the inhibition of VEGF/VEGFR. Furthermore, genomic instability would enable cancer cells to bypass the VEGF/VEGFR axis and stimulate new blood vessel growth using alternative signaling pathways. Future therapy should focus on targeting molecules, as well as VEGF, that are present on large blood vessels’ EC lining. Targeting large vessels could stop the blood perfusion to the entire mass of the tumour, hence this would enhance the pruning of microvessels that are sensitive to the inhibition of VEGF/VEGFR. This concept was tested and supported by the findings of a study that utilised photodynamic energy to thrombose and subsequently blocked the main arteries and draining veins of a mouse ear tumour [103].

Another novel strategy is the use of nano-particles to deliver specific anti-angiogenic agents [104]. For example, endostatin, a protein that was extracted for the first time in 1996 from murine hemagioendothelioma (EOMA) cell culture medium [32]. Endostatin has a potent anti-angiogenic effect. The exact molecular anti-angiogenic mechanism(s) of endostatin are not fully understood and subject to investigation. In vitro and vivo studies showed that endostatins induce endothelial cell apoptosis, and suppress its proliferation and migration via a complex network of signaling [105]. However, there are important challenges in the clinical application of endostatin related to the chemical nature of this protein. These challenges include the short half-life and instability of the protein in vivo [106], the requirement of administering high volumes of endostatin to exert their anti-angiogenic effects, which in itself is associated with significant practical and cost implications [107], as well as technological challenges related to manufacturing a correctly folded and soluble protein to ensure adequate bioactivity within the tumour cells [108]. To overcome these challenges, nanotechnology has been utilized to manufacture nanoparticles as transporters of this protein [109]. Cancer cells are shown to readily uptake nano-particles, and therefore the anti-tumoural activity of endostatin is enhanced when delivered via nano-particles [110]. In addition, by adding nine amino acids to the N-terminal of recombinant human endostain, endostar is produced. Endostar is a more stable molecular bioengineered form of endostatin. This is because endostar is better at resisting degradation by proteolytic enzymes and more stable during temperature changes [111].

Two independent studies by Chen et al. [112] and Hu et al. [113] have confirmed that endostar carried by nanoparticles have a better anticancer activity than the conventional delivery system because of the improved release and longer half-life of endostar in target tumour. Chen et al. studied prepared particulate carriers (nanoparticles and microspheres) of poly (DL-lactide-co-glycolide) (PLGA) and poly (ethylene glycol) (PEG)-modified PLGA (PEG-PLGA) to promote a better delivery and release of endostar, as the nano-transporter enables high encapsulation, rapidly release and the higher cancer intracellular bioavailability of endostar.

As explained above, the vessel co-option acts as an important mechanism of resistance to anti-angiogenesis as well as an important source of blood supply that supports the growth of tumors. Therefore, the inhibition of the vessel co-option is the focus of many research groups, through targeting cell motility or adhesion pathways in tumour stroma. In in a mouse model of liver metastases, Frentzas et al. [60] showed that, by silencing the expression of actin-related protein 2/3 (ARP2/3), a protein complex involved in actin-mediated cell motility, and the vessel co-option, can be inhibited. Interestingly, preclinical trials showed improved tumour control when VEGF and vessel co-options are inhibited simultaneously compared to the blocking of VEGF signalling alone [114]. Another novel approach that has been tested in mouse models of brain-metastatic breast cancer and showed some promising results, is the inhibition of the adhesion of cancer cells to pre-existing blood vessels to block vessel co-option through inhibition of L1 Cell Adhesion Molecule (L1CAM) or the cell adhesion receptor β1 integrin [115]. Moreover, pre-clinical models of glioma, and metastases to the liver, lymph nodes or lungs that are vessel co-option-dependent, showed that blocking both the angiopoietin and VEGF pathways was more effective compared to the inhibition of VEGF alone [116,117]. The exact role of angiopoietin in the recruitment or maintenance of co-opted tumour vessels is not fully understood [118]. However, a phase 2 clinical trial of angiopoietin inhibition with and without bevacizumab in recurrent glioblastoma did not show any improvement in progression free survival (PFS) [118].

## 10. Conclusions

Excessive, insufficient or abnormal angiogenesis contributes to tumour survival, growth invasion and metastasis. Targeting single angiogenic (pro or inhibitory) molecules showed promising results in animal trials, but has been of limited success in human cancer. To date, despite their modest impact, anti VEGF continues to be one of the treatment lines of several solid malignancies. Nevertheless, it is believed that antiangiogenic monotherapy aiming at single molecule activity is insufficient to combat the myriad of angiogenic factors produced by cancer cells and its microenvironment and this would explain, at least partly, the modest effect of anti VEGF strategies. Future challenges include a detailed understanding of the many angio-modulating pathways in a more integrated manner to identify more holistic therapeutic approaches to improve survival rate in cancer patients.

## Figures and Tables

**Figure 1 cancers-12-01172-f001:**
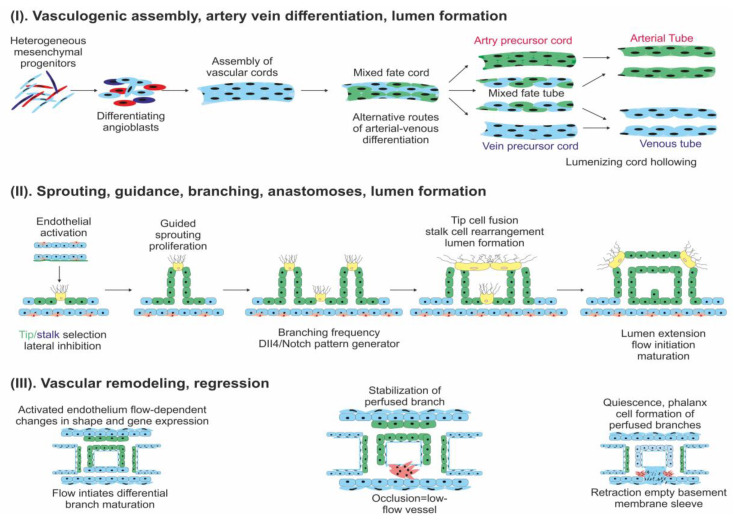
Steps of angiogenesis: (**I**)—Endothelial cell (EC) differentiated from angioblasts. (**II**)—sprouting, guidance, branching, anastomoses, lumen formation. (**III**)—vascular remodeling from a primitive (left box) towards a stabilized and mature vascular plexus (right box).

**Figure 2 cancers-12-01172-f002:**
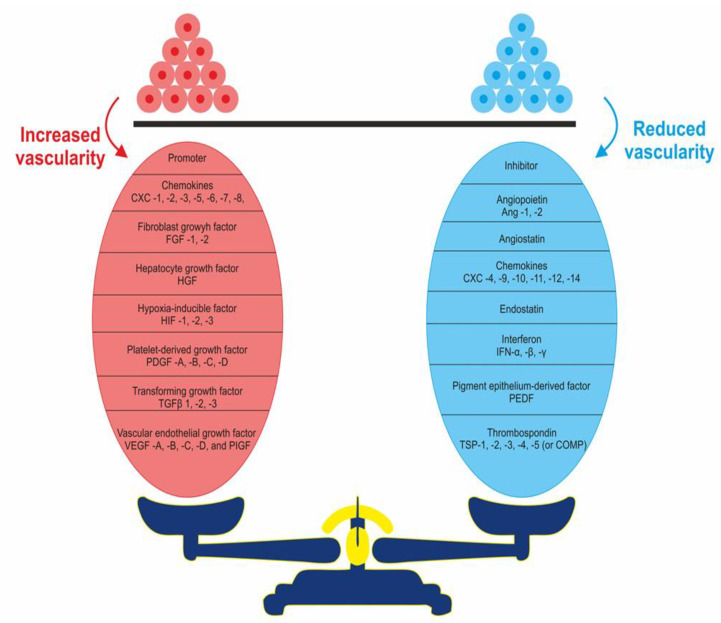
Maintaining homeostasis results from an equilibrium between promotors and inhibitors of angiogenesis.

**Figure 3 cancers-12-01172-f003:**
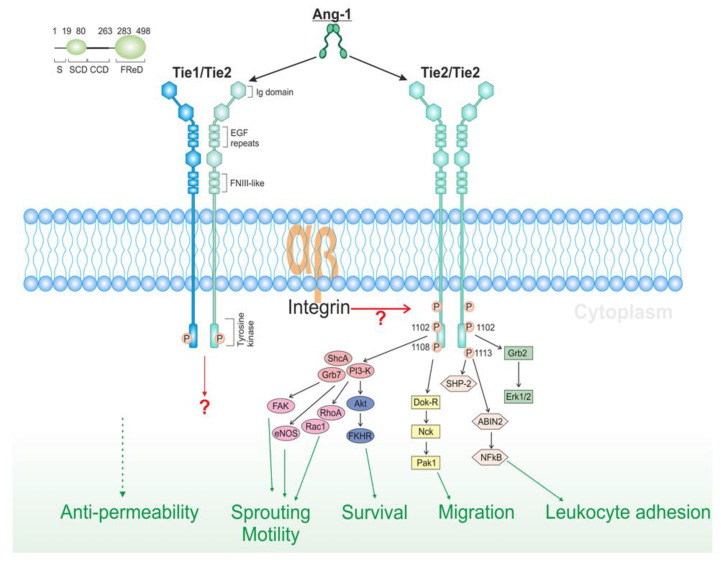
Ang1 and Ang2 bind to Tie2 with similar affinities; however, whereas Ang1 is an agonist, the ability of Ang2 to activate Tie2 appears to depend on the cell type and context. The activation of the Tie2 pathway results in the inhibition of apoptosis, cell survival and migration.

**Figure 4 cancers-12-01172-f004:**
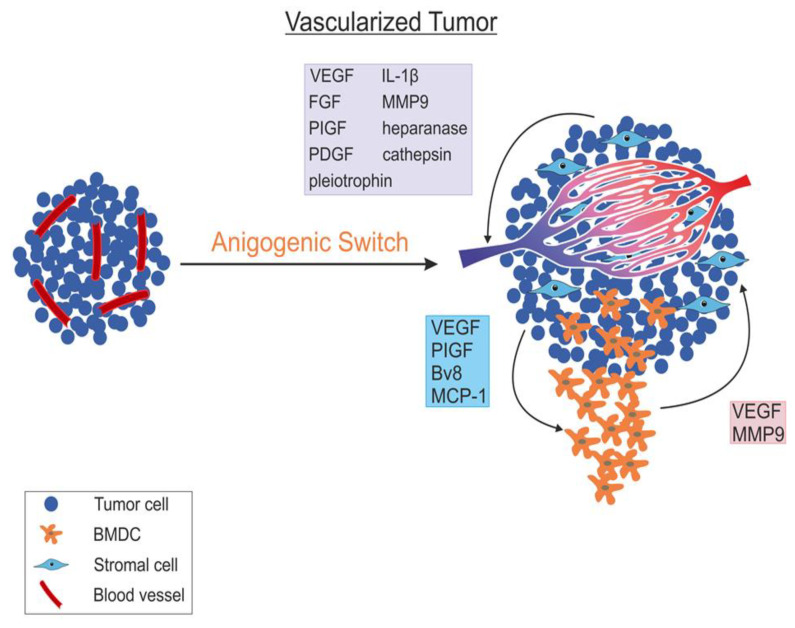
Angiogenic switch in transgenic mouse, showing progression from hyperplasia to hyper-vascularised tumour. The pro-angiogenic factors and proteases secreted by the tumour cells themselves (green box) and the cells of the immune system recruited to the tumour site (pink box), and the factors secreted by the tumour cells to recruit inflammatory cells (blue box).

**Figure 5 cancers-12-01172-f005:**
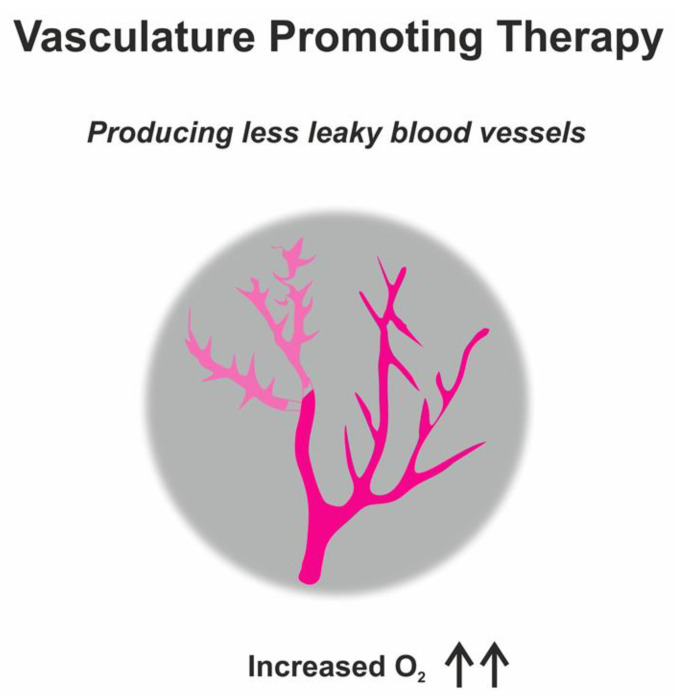
Promoting tumour blood supply to improve cytotoxic delivery to tumour. This approach might be particularly effective in tumours that are poorly supplied by blood, such as pancreatic cancer.

**Figure 6 cancers-12-01172-f006:**
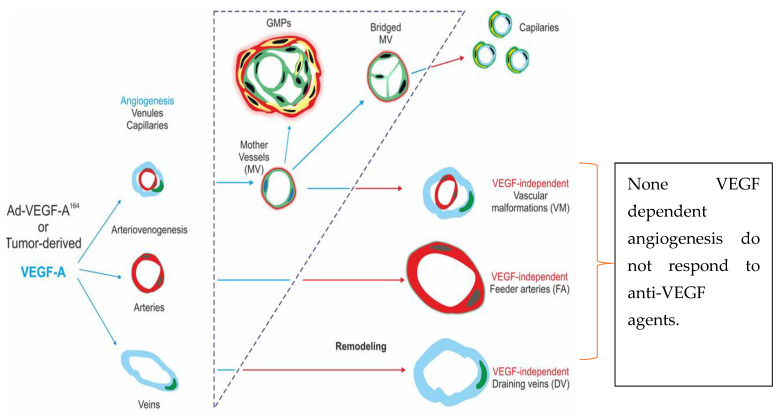
VEGF-A plays an important role in angiogenesis. The inhibition of VEGF-A prevents new vessel formation. VEGF-independent angiogenesis are not sensitive to the inhibition of VEGF-A.

**Table 1 cancers-12-01172-t001:** AngiomiR are none-coding RNAs that play an important role in angiogenesis in normal tissue, through their expression or silencing depending on physiological demand. The dysregulation of miR-200b is detected in some cancers. Different types of AngiomiR have specific effects on angiogenesis.

AngiomiR	Molecular Function	Reference
miR-15b, miR-16, miR-20a, miR-20b	Have no known functions. They might contribute in regulation of VEGF.	[10]
miR-21, miR-31	Triggers mobilisation of EC.	[11]
miR-17-92	Dysregulation of miR-17-92 in cancer cells promote growth.	[12]
miR-130a	Induces angiogenesis by supressing GAX and HOXA5	[13]
miR-296	Animal studies showed that by acting on HGS, miR-296 stimulate angiogenesis.	[14]
miR-320	Suppression of miR-320 in diabetic cells trigger angiogenesis by stimulating EC proliferation.	[15]
miR-210	In hypoxic cell culture, miR-210 promote EC proliferation and survival.	[16]
miR-378	Support tumour growth by improving vascularisation via angiogenesis.	[17]

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
