# Peer review of "Inducing Angiogenesis, a Key Step in Cancer Vascularization, and Treatment Approaches"

_cancers, 2020, doi:10.3390/cancers12051172_

Round 1

Reviewer 1 Report

This manuscript from Saman et al reviews current understanding of different pathways of angiogenesis and current treatment options. The section reviewing treatments is particularly strong. Multiple points should be addressed to strengthen this review to be well understood by outside readers.

  1. The title should be corrected from “Hall Marks” to Hallmark
  2. In lines 34-37, the statement regarding angiogenesis within the tumor vs in culture is not clear. Are the tumor cells part of an organ culture? It is not clear whether there is angiogenesis occurring or the culture conditions just have much higher levels of oxygen. The reference that is cited is a review article, and it would be beneficial to cite the original study.
  3. Figure 1. The figure legend has A, B, and C referenced in the figure legend, but the sections within the figure are marked by I, II, III.
  4. Lines 49-50 contain a paragraph break that should not be there.
  5. In section 2, the only reference cited multiple times is a different review article. This section should be revised to contain material from other sources.
  6. In line 84-86, the sentence is missing some words. “initiation of tumor.”? This should be clarified.
  7. Decreasing some of the abbreviations could help with clarity. A number of the abbreviated words are not included in the abbreviations section.
  8. In lines 175, 197, and 200, “trail” should be corrected to trial.
  9. It would be helpful to provide more explanation in the text to support Figure 6. The terms mother vessels, bridged vessels, GMPs?, vascular malformations, feeder arteries, draining vessels are not defined in the text. Does VEGF-A therapy have different impacts on each of these types of vessels? These outcomes are not clear from the text.
  10. Secret should be corrected to secrete multiple times in the text.
  11. The text should be edited to correct repeated references (one in brackets then a repeat in parenthesis), and hyphens are not consistent in abbreviations.

Author Response

Revivewers#1

This manuscript from Saman et al reviews current understanding of different pathways of angiogenesis and current treatment options. The section reviewing treatments is particularly strong. Multiple points should be addressed to strengthen this review to be well understood by outside readers.

Authors Response (AR): We are very thankful to the reviewer for raising very valuable suggestions. We have included all the issues raised by the esteemed reviewer in the revised manuscript.

  1. The title should be corrected from “Hall Marks” to Hallmark

AR:  We have given a better and more appropriate title for this article  as it has been ask by reviewer 3. Now it reads as “ Inducing angiogenesis, a key step in cancer vascularization, and treatment approaches”

  1. In lines 34-37, the statement regarding angiogenesis within the tumor vs in culture is not clear. Are the tumor cells part of an organ culture? It is not clear whether there is angiogenesis occurring or the culture conditions just have much higher levels of oxygen. The reference that is cited is a review article, and it would be beneficial to cite the original study.

AR:  As per reviewer suggestion we have made the following addition in the revised manuscript:

“The continued growth of cancer cells in angiogenesis rich cell culture is explained by reproducing physiological properties of a three-dimensional cell culture model that provides controlled fluid perfusion that permits the regulation of oxygen intake, promoting a circulatory environment that is controlled by computer hardware”

  1. Figure 1. The figure legend has A, B, and C referenced in the figure legend, but the sections within the figure are marked by I, II, III.

AR:  As per reviewer suggestion correction to figure 1legend made to I, II, III

  1. Lines 49-50 contain a paragraph break that should not be there.

AR: The paragraph break removed.

  1. In section 2, the only reference cited multiple times is a different review article. This section should be revised to contain material from other sources.

AR: based on the above feedback section was expanded to include the following paragraph and an additional Table 1 was added in the revised manuscript.

“At a molecular level, angiogenesis is normally controlled by a family of small none coding RNA molecules that are collectively called angiomiRs 25761972. AngiomiRs comprised of pro-angiogenic miRs and anti-angiogenic miRs (Table 1)[6]. A well-studied angiomiRs is miR-200b, which belongs to the miR-200 family [7]. miR-200b has antiangiogenic effect. Its expression is transiently turned down when new vessel formation is required, for example during wound healing, once the physiological demand subsided, miR-200b is expressed again to stop angiogenesis as a measure of tight control on new vessel formation. Downregulation of miR-200b in response to tissue hypoxia triggers epithelial to mesenchymal transition and modulating endothelial cell migration which result in new vessel formation[8]. There is evidence that dysregulation of iR-200b contribute to oncogenesis and metastasis in some cancers such as breast cancer[9]”.

  1. In line 84-86, the sentence is missing some words. “initiation of tumor.”? This should be clarified.

AR:  As per reviewer suggestion we have added   the following to update the role of VEGF in tumour initiation:

Upregulation VEGF initiate tumorigenesis by contributing to the activation of epithelial– mesenchymal transition (EMT) [17]. EMT represents a key event in the process of new vessel formation. [18]. This  because EMT leads to loss of cell polarity, dramatic cytoskeletal changes, which lead to increased cell motility and loss cell to cell adhesion by the loss of E-cadherin and ZO-1. The last two markers are associated with epithelial cells. EMT also results in production of several proteolytic enzymes, including matrix metalloproteases and serine proteases that degrade extracellular matrix (ECM). Several pathways involved in EMT supporting endothelial cell (EC) survival and proliferation [18]. These pathways invlove complex interactions between cell membrane, ECM and intracellular regulatory signalling pathways. The resulting phenotypical changes caused by EMT promote cancer cell invasion of basement membrane and eventually cancer cell metastasis [19].

  1. Decreasing some of the abbreviations could help with clarity. A number of the abbreviated words are not included in the abbreviations section.

AR: As per suggestion we have  reviewed  this point and a few them were removed.

  1. In lines 175, 197, and 200, “trail” should be corrected to trial.

AR:  It has been corrected in the revised manuscript

  1. It would be helpful to provide more explanation in the text to support Figure 6. The terms mother vessels, bridged vessels, GMPs?, vascular malformations, feeder arteries, draining vessels are not defined in the text. Does VEGF-A therapy have different impacts on each of these types of vessels? These outcomes are not clear from the text.

AR: In response to comment 9, the following was added in the revised manuscript

Several studies showed that arteriol formation and tortuosity as well as venous dilation are increased through VEGF expression [64]. Cell culture injected with adenovirus expressing VEGF undergo induction of mother vessels (MV) and stabilized MV from normal capillaries and venules. In contrast inhibition of VEGF is shown to cause veins and arterioles to have fewer cleavage planes. For example, Aflibercept, a decoy receptor that binds VEGF-A, induces rapid collapse of mother vessels (MV) to glomeruloid microvascular proliferations (GMP). VEGF inhibition, by anti-VEGF/VEGF receptor, is shown to restore vasculature within hours to normal microvessels by way of GMP [65]. GMP is believed to act as an intermediary step in MV reversion to normal microvessels after VEGF blockade[66],

  1. Secret should be corrected to secrete multiple times in the text.

AR: It has been corrected in the revised manuscript.

  1. The text should be edited to correct repeated references (one in brackets then a repeat in parenthesis), and hyphens are not consistent in abbreviations.

AR: We appreciate for pointing out this and it has been corrected in the revised manuscript.

Reviewer 2 Report

The review titled "Inducing Angiogenesis, a Hall Marks of Cancer, and Treatment Approaches" by Saman et al. it is correctly set up and with sufficiently updated indications on the state of the art regarding tumor angiogenesis and part of the clinical treatment in subjects suffering from solid tumors.
However, in the introductory part there is absolutely no reference to the epithelium / mesenchyme transition which is the basis of both normal and etumoral angiogenesis. Here are some articles that could be cited in order to support this part.
- Ski Rep. 2016 Dec 2; 6: 38221. doi: 10.1038 / srep38221.
Dynamic Microenvironment Induces Phenotypic Plasticity of Esophageal Cancer Cells Under Flow.
Calibasi Kocal G, Güven S, Foygel K, Goldman A, Chen P, Sengupta S, Paulmurugan R, Baskin Y, Demostri U.
- J Oral Pathol Med. 2014 Apr; 43 (4): 250-7. doi: 10.1111 / jop.12116. Epub 2013 Sep 11.
Cetuximab inhibits oral squamous cell carcinoma invasion and metastasis via degradation of epidermal growth factor receptor.
Dai W, Li Y, Zhou Q, Xu Z, Sun C, Tan X, Lu L.
- Front Biosci. 2008 Jan 1; 13: 2335-55.
Roles of molecules involved in epithelial / mesenchymal transition during angiogenesis.
Ghersi G.
Furthermore, in the part relating to the pharmacological treatment of situations in which there is an important structural angiogenesis such as solid tumors and retinopathies, no mention is made of the use of nano-transporters / nano-vectors capable of specifically localizing the effect of the drug to to the detriment of the side effects due to chemotherapy, a process that sees the use of the drug in less specific ways. Also in this sense we can suggest some works that well describe these phenomena, reported below:
- J Cell Physiol. 2018 Apr; 233 (4): 2902-2910. doi: 10.1002 / jcp.26029. Epub 2017 Jun 22.
Nanoparticles as new tools for inhibition of cancer angiogenesis.
Hashemi Goradel N, Ghiyami-Hour F, Jahangiri S, Negahdari B, Sahebkar A, Masoudifar A, Mirzaei H.
- Adv Pharm Bull. 2017 Apr; 7 (1): 21-34. doi: 10.15171 / apb.2017.004. Epub 2017 Apr 13.
The Challenges of Recombinant Endostatin in Clinical Application: Focus on the Different Expression Systems and Molecular Bioengineering.
Mohajeri A, Sanaei S, Kiafar F, Fattahi A, Khalili M, Zarghami N
- Expert Opin Drug Deliv. 2011 Aug; 8 (8): 1041-56. doi: 10.1517 / 17425247.2011.585155. Epub 2011 May 17.
Antiangiogenic anticancer strategy based on nanoparticulate systems.
Yoncheva K, Momekov G.
- Nanomaterials (Basel). 2020 Mar 22; 10 (3). pii: E581. doi: 10.3390 / nano10030581.
High Density Display of an Anti-Angiogenic Peptide on Micelle Surfaces Enhances Their Inhibition of αvβ3 Integrin-Mediated Neovascularization In Vitro.
Nagaraj R, Stack T, Yi S, Mathew B, Shull KR, Scott EA, Mathew MT, Bijukumar DR.
- Int J Ophthalmol. 2018 Jun 18; 11 (6): 1038-1044. doi: 10.18240 / ijo.2018.06.23. eCollection 2018.
Nanotechnology in retinal drug delivery.
Jiang S, Franco YL, Zhou Y, Chen J.
- Eur J Pharm Biopharm. 2017 Aug; 117: 385-399. doi: 10.1016 / j.ejpb.2017.05.005. Epub 2017 May 13.
Novel inulin-based mucoadhesive micelles loaded with corticosteroids as potential transcorneal permeation enhancers.
Di Prima G, Saladino S, Bongiovì F, Adamo G, Ghersi G, Pitarresi G, Giammona G
- J Control Release. 2017 Feb 28; 248: 96-116. doi: 10.1016 / j.jconrel.2017.01.012. Epub 2017 Jan 11.
Polymeric micelles for ocular drug delivery: From structural frameworks to recent preclinical studies.
Mandal A, Bisht R, Rupenthal ID, Mitra AK.
It is hoped that these suggestions will strengthen the content of the review and give it a better and more modern reading.

Author Response

Reviewer#2:

Comments and Suggestions for Authors

The review titled "Inducing Angiogenesis, a Hall Marks of Cancer, and Treatment Approaches" by Saman et al. it is correctly set up and with sufficiently updated indications on the state of the art regarding tumor angiogenesis and part of the clinical treatment in subjects suffering from solid tumors.

AR: We are very thankful to reviewer for a positive feedback. We also appreciate for mentioning that “it is correctly set up and with sufficiently updated indications on the state of the art regarding tumor angiogenesis and part of the clinical treatment in subjects suffering from solid tumors”.

We have replied all the issue raised point by point as follows and incorporated all the suggestions in the revised manuscript.

However, in the introductory part there is absolutely no reference to the epithelium / mesenchyme transition which is the basis of both normal and etumoral angiogenesis. Here are some articles that could be cited in order to support this part.
- Ski Rep. 2016 Dec 2; 6: 38221. doi: 10.1038 / srep38221.
Dynamic Microenvironment Induces Phenotypic Plasticity of Esophageal Cancer Cells Under Flow.
Calibasi Kocal G, Güven S, Foygel K, Goldman A, Chen P, Sengupta S, Paulmurugan R, Baskin Y, Demostri U.
- J Oral Pathol Med. 2014 Apr; 43 (4): 250-7. doi: 10.1111 / jop.12116. Epub 2013 Sep 11.
Cetuximab inhibits oral squamous cell carcinoma invasion and metastasis via degradation of epidermal growth factor receptor.
Dai W, Li Y, Zhou Q, Xu Z, Sun C, Tan X, Lu L.
- Front Biosci. 2008 Jan 1; 13: 2335-55.
Roles of molecules involved in epithelial / mesenchymal transition during angiogenesis.
Ghersi G.

AR:  We are very thankful for raising this important point . We have updated this section with following information in the revised manuscript

Upregulation VEGF initiate tumorigenesis by contributing to the activation of epithelial– mesenchymal transition (EMT) [17]. EMT represents a key event in the process of new vessel formation. [18]. This  because EMT leads to loss of cell polarity, dramatic cytoskeletal changes, which lead to increased cell motility and loss cell to cell adhesion by the loss of E-cadherin and ZO-1. The last two markers are associated with epithelial cells. EMT also results in production of several proteolytic enzymes, including matrix metalloproteases and serine proteases that degrade extracellular matrix (ECM). Several pathways involved in EMT supporting endothelial cell (EC) survival and proliferation [18]. These pathways invlove complex interactions between cell membrane, ECM and intracellular regulatory signalling pathways. The resulting phenotypical changes caused by EMT promote cancer cell invasion of basement membrane and eventually cancer cell metastasis [19].

Furthermore, in the part relating to the pharmacological treatment of situations in which there is an important structural angiogenesis such as solid tumors and retinopathies, no mention is made of the use of nano-transporters / nano-vectors capable of specifically localizing the effect of the drug to to the detriment of the side effects due to chemotherapy, a process that sees the use of the drug in less specific ways. Also in this sense we can suggest some works that well describe these phenomena, reported below:
- J Cell Physiol. 2018 Apr; 233 (4): 2902-2910. doi: 10.1002 / jcp.26029. Epub 2017 Jun 22.
Nanoparticles as new tools for inhibition of cancer angiogenesis.
Hashemi Goradel N, Ghiyami-Hour F, Jahangiri S, Negahdari B, Sahebkar A, Masoudifar A, Mirzaei H.
- Adv Pharm Bull. 2017 Apr; 7 (1): 21-34. doi: 10.15171 / apb.2017.004. Epub 2017 Apr 13.
The Challenges of Recombinant Endostatin in Clinical Application: Focus on the Different Expression Systems and Molecular Bioengineering.
Mohajeri A, Sanaei S, Kiafar F, Fattahi A, Khalili M, Zarghami N
- Expert Opin Drug Deliv. 2011 Aug; 8 (8): 1041-56. doi: 10.1517 / 17425247.2011.585155. Epub 2011 May 17.
Antiangiogenic anticancer strategy based on nanoparticulate systems.
Yoncheva K, Momekov G.
- Nanomaterials (Basel). 2020 Mar 22; 10 (3). pii: E581. doi: 10.3390 / nano10030581.
High Density Display of an Anti-Angiogenic Peptide on Micelle Surfaces Enhances Their Inhibition of αvβ3 Integrin-Mediated Neovascularization In Vitro.
Nagaraj R, Stack T, Yi S, Mathew B, Shull KR, Scott EA, Mathew MT, Bijukumar DR.
- Int J Ophthalmol. 2018 Jun 18; 11 (6): 1038-1044. doi: 10.18240 / ijo.2018.06.23. eCollection 2018.
Nanotechnology in retinal drug delivery.
Jiang S, Franco YL, Zhou Y, Chen J.
- Eur J Pharm Biopharm. 2017 Aug; 117: 385-399. doi: 10.1016 / j.ejpb.2017.05.005. Epub 2017 May 13.
Novel inulin-based mucoadhesive micelles loaded with corticosteroids as potential transcorneal permeation enhancers.
Di Prima G, Saladino S, Bongiovì F, Adamo G, Ghersi G, Pitarresi G, Giammona G
- J Control Release. 2017 Feb 28; 248: 96-116. doi: 10.1016 / j.jconrel.2017.01.012. Epub 2017 Jan 11.
Polymeric micelles for ocular drug delivery: From structural frameworks to recent preclinical studies.
Mandal A, Bisht R, Rupenthal ID, Mitra AK.
It is hoped that these suggestions will strengthen the content of the review and give it a better and more modern reading.

AR:  We have updated as per reviewer’s suggestion the following under Novel therapies with references.

“Another novel strategy is the use of nano-particles to deliver specific anti-angiogenic agents [96]. For example, endostatin, a protein that was extracted for the first time in 1996 from murine hemagioendothelioma (EOMA) cell culture medium [24]. Endostatin has a potent anti-angiogenic effect. The exact molecular anti-angiogenic mechanism(s) of endostatin is not fully understood and is a subject to investigation. In vitro and vivo studies showed that endostatins induce endothelial cell apoptosis, suppress its proliferation and migration via a complex network of signaling [97]. However, there are important challenges in the clinical application of endostatin related to the chemical nature of this protein. These challenges include the short half-life and instability of the protein in vivo [98], the requirement of administering high volume of endostatin to exert its anti-angiogenic effect which in itself is associated with significant practical and cost implications [99], as well as technological challenges related to manufacturing a correctly folded and soluble protein to ensure adequate bioactivity within the tumour cells [100]. To overcome these challenges nanotechnology has been utilized to manufacture nanoparticles as transporters of this protein [101]. Cancer cells are shown to readily uptake nano-particles therefore enhancement of anti-tumor activity of endostatin when delivered via nano-particles[102]. In addition, by adding 9 amino acids to N-terminal of recombinant human endostain, endostar is produced. Endostar is a more stable molecular bioengineered form of endostatin. This because endostar is better at resisting degradation by proteolytic enzymes and more stable during temperature changes[103].

Two independent studies by Chen et al. [104]and Hu et al [105] have confirmed that endostar carried by nanoparticles have a better anticancer activity than conventional delivery system because of improved releasing and longer half –life of endostar in target tumor. Chen et al studied prepared particulate carriers (nanoparticles and microspheres) of poly (DL-lactide-co-glycolide) (PLGA) and poly (ethylene glycol) (PEG)-modified PLGA (PEG-PLGA) to promote a better delivery and release of endostar as the nano-transporter enables high encapsulation, rapidly releasing and higher cancer intracellular bioavailability of endostar.

Reviewer 3 Report

Overall this is a well organised review on angiogenesis that covers the basic aspects and is an useful reading for new comers to the topic.
The main problem is with the discussion of role played by angiogenesis in cancer and its place in the broader field of the relationship between cancer and blood vessels.

Since the mid 1990s has emerged that tumours can also growth, progress and metastatise  in a non-angiogenic fashion. The current definition of “inducing angiogenesis” as an Hallmark of cancer is incorrect and revision is under way: for recent overviews on the topic see:

Donnem et al. Non-angiogenic tumours and their influence on cancer biology Nature Reviews Cancer 18:323, 2018.

Kuczynski et al. Vessel Co-option Nature Review Clinical Oncology 16:469, 2019

By now an extensive literature is also available.

Title

It should be changed to reflect the fact that angiogenesis is only one of the means of cancer vascularization

Introduction.

“ in the absence of angiogenesis, tumour could grow to a maximum of 1-2 mm3 in diameter before they stopped growing  and died.”

This is an example of the concepts which are no longer valid (see above reviews).

Angiogenesis in cancer

This paragraph needs to be reviewed and should reflects the fact that angiogenesis is now only one of the ways cancers are vascularised

Angiogenic switch.

One of the new issued raised by the discovery of non-angiogenic tumours is why some tumours switch to angiogenesis and some do not (but still grow), and how the tumours can switch back from angiogenic to non angiogenic e.g. after anti angiogenic treatment (e.g. Kuczynski, E. A. et al. Co‑option of liver vessels and not sprouting angiogenesis drives acquired sorafenib resistance in hepatocellular carcinoma. J. Natl. Cancer Inst. 108, djw030 (2016)). Some candidate switch genes have been proposed (e.g. Auf, G. et al. Inositol-requiring enzyme 1alpha is a key regulator of angiogenesis and invasion in malignant glioma. Proc. Natl Acad. Sci. USA 107, 15553–15558 (2010)).

Anti angiogenic treatment

The authors should discuss vascular co-option as escape mechanism (see:

 Kuczynski et al. Vessel Co-option Nature Review Clinical Oncology 16:469, 2019; 

Donnem et al Vessel co‐option in primary human tumors and metastases: an obstacle to effective anti‐angiogenic treatment? Cancer medicine 2 (4), 427-436

Kuczynski and Reynolds Vessel co-option and resistance to anti-angiogenic therapy

Angiogenesis 23, pages55–74(2020).

Even in animal model tumour progression during Bevacizumab treatment ha been described (e.g. Leenders et al. Antiangiogenic therapy of cerebral melanoma metastases results in sustained tumor progression via vessel co-option. Clinical Cancer Research 10:6222  2004)

Figure 6 is not clear. It should be   better explained

Author Response

Reviewer#3

Comments and Suggestions for Authors

Overall this is a well organised review on angiogenesis that covers the basic aspects and is an useful reading for new comers to the topic.
The main problem is with the discussion of role played by angiogenesis in cancer and its place in the broader field of the relationship between cancer and blood vessels.

  ince the mid 1990s has emerged that tumours can also growth, progress and metastatise  in a non-angiogenic fashion. The current definition of “inducing angiogenesis” as an Hallmark of cancer is incorrect and revision is under way: for recent overviews on the topic see:

Donnem et al. Non-angiogenic tumours and their influence on cancer biology Nature Reviews Cancer 18:323, 2018.

Kuczynski et al. Vessel Co-option Nature Review Clinical Oncology 16:469, 2019

By now an extensive literature is also available.

AR:  We are very thankful to the reviewer for a positive feedback. We are appreciation for mentioning that “Overall this is a well-organized review on angiogenesis that covers the basic aspects and is an useful reading for new comers to the topic”

We have made following changes as per reviewer suggestions.

The abstract now reads as;

Angiogenesis is a term that describes the formation of new blood and lymphatic vessels from a pre-existing vasculature. This allows tumor cells to acquire sustenance in the form of nutrients and oxygen and the ability to evacuate metabolic wastes. As one of the hallmarks of cancer, angiogenesis has been studied extensively in animal and human models to enable better understanding of cancer biology and development of new anti-cancer treatment. Angiogenesis plays a crucial role in the process of tumour-genesis, because solid tumors need a blood supply if they are to grow beyond a few millimetres in size. On the other hands, there is growing evidence that some solid tumors exploit existing normal blood supply and bypass new vessel formation to grow. This review of literature will present the current understanding of this intricate process and the latest advances in the use of angiogenesis targeting therapies in the fight against cancer.

Title

It should be changed to reflect the fact that angiogenesis is only one of the means of cancer vascularization

 AR: We agree with reviewer and  the title changed to: “Inducing angiogenesis, a key step in cancer vascularization, and treatment approaches” in the revised manuscript

Introduction.

 “ in the absence of angiogenesis, tumour could grow to a maximum of 1-2 mm3 in diameter before they stopped growing  and died.”

This is an example of the concepts which are no longer valid (see above reviews).

 AR: The following addition is made to explain and expand on non-angiogenic (vessel co-option), which also reflects on the next two comments from the third reviewer:

“Importantly, not all tumors rely on new blood vessel formation to survive and grow [39] and therefore the angiogenic switch might never occur. Some tumors exploit existing blood supply through a process named: vessel co-option to support their growth and to enable metastasis. Vessel co-option has been observed in a number of tumors such as non-small cell lung cancer (NSCLC), glioblastoma and hepatocellular carcinoma[40,41]. Cancer cells seem to grow along existing vessels and/or invade the connective tissue that present between the vessels allowing the cancer cells to incorporate to the existing normal vasculature to begin hijacking the blood supply [42,43]. There is evidence that vessel co-option promotes cancer cell motility and metastasis and tumour dormancy[44]. Moreover, some tumours such as NSCLC, use both angiogenesis and vessel co-option simultaneously or sequentially ( in no particular order) to acquire blood supply and venous and lymphatic drainage [45]. Moreover, there is growing evidence that increased vascularity, often measured through microvascular density, caused by vessel co-option ,is associated with higher tumour grade and higher risk of metastasis[46]. Interestingly bone marrow appears to be an important site for vessel co-option in both primary and secondary bone malignancies which in turn might explain the development of tumour dormancy in bones and higher rate of chemoresistance[47,48]”.

Angiogenesis in cancer

This paragraph needs to be reviewed and should reflects the fact that angiogenesis is now only one of the ways cancers are vascularised

 AR: We have included this suggestion in e above response:

Angiogenic switch.

One of the new issued raised by the discovery of non-angiogenic tumours is why some tumours switch to angiogenesis and some do not (but still grow), and how the tumours can switch back from angiogenic to non angiogenic e.g. after anti angiogenic treatment (e.g. Kuczynski, E. A. et al. Co‑option of liver vessels and not sprouting angiogenesis drives acquired sorafenib resistance in hepatocellular carcinoma. J. Natl. Cancer Inst. 108, djw030 (2016)). Some candidate switch genes have been proposed (e.g. Auf, G. et al. Inositol-requiring enzyme 1alpha is a key regulator of angiogenesis and invasion in malignant glioma. Proc. Natl Acad. Sci. USA 107, 15553–15558 (2010)).

 AR:  We have made following addition in the revised manuscript to update this issue.

“In addition, there is also evidence from preclinical studies that show some tumours such NSCLC and gliomas never undergo angiogenic switch and rely only on vessel co-option[45,49]. In contrast some tumours for example hepatocellular carcinoma and liver metastases of gastrointestinal tract switch from using vessel co-option at early stages of tumorigenesis to angiogenesis at a later stage during tumour progression [50,51]. Albeit, this progression from vessel co-option to angiogenesis is not an obligatory requirement of tumour progression and metastasis [52]. Moreover preclinical and clinical studies showed that there are at times, but not always, differences exist between primary and secondary of the same tumour in term of their access to blood supply [53,54]. For example, when cells from angiogenic primary human breast tumours spread to the lung tissue, they switch to vessel co-option as a mode of accessing blood supply [54] and which also functions as resistant mechanism against antiangiogenic therapy [46].

Anti angiogenic treatment

The authors should discuss vascular co-option as escape mechanism (see:

 Kuczynski et al. Vessel Co-option Nature Review Clinical Oncology 16:469, 2019; 

Donnem et al Vessel co‐option in primary human tumors and metastases: an obstacle to effective anti‐angiogenic treatment? Cancer medicine 2 (4), 427-436

Kuczynski and Reynolds Vessel co-option and resistance to anti-angiogenic therapy

Angiogenesis 23, pages55–74(2020).

 AR:  We appreciate reviewer for this valuable suggestion  and  added following updates in the revised manuscript.

“Vessels co-option as a mechanism to attain blood supply by cancer cells is another resistant mechanism to anti-angiogenic treatment. Preclinical models demonstrated switch from angiogenesis to vessel co-option during anti-angiogenic treatment [90,91]. Escaping anti-angiogenic agents’ effect by using vessel co-option is seen across a range of cancer types. For example, the modest response of glioma to bevacizumab is shown, in preclinical studies and clinical case reports, to be due to vessel co-option [92]. This could be intrinsic resistance or acquired during treatment with bevacizumab due to switch from angiogenesis to vessel co-option [93,94]. This switch from angiogenesis to vessel co-option is also observed during treatment of breast cancer with anti-angiogenic therapy. Pulmonary metastasis from breast cancer are shown to use lung parenchymal blood supply for their survival and growth which explains their resistance to anti-angiogenic therapy [53]. In addition preclinical trial showed that after an initial response of xenograft model of hepatocellular carcinoma to sorafenib ( a multi-kinase inhibitor with antiangiogenic properties), the tumour progressed within a month due to large-scale co-option of sinusoidal and portal tract vessels[46]  Moreover, several studies showed that resistance to anti-angiogenic therapy in metastatic colorectal carcinoma (CRC) to the liver is likely secondary to CRC cells co-option of pre-existing liver vessels; this can occur in the context of both intrinsic and acquired resistance[52]”.

Even in animal model tumour progression during Bevacizumab treatment ha been described (e.g. Leenders et al. Antiangiogenic therapy of cerebral melanoma metastases results in sustained tumor progression via vessel co-option. Clinical Cancer Research 10:6222  2004)

 AR:  We have provided in the above response examples of xenograft models. In addition, following  novel approaches updated in the revised manuscript  to reflect on the role of vessel co-option and therapeutic opportunity of its inhibition:

“As explained above vessel co-option act as an important mechanism of resistance to anti-angiogenesis as well as an important source of blood supply that support the growth of tumour. Therefore, inhibition of vessel co-option is the focus of many research groups, through targeting cell motility or adhesion pathways in tumour stroma. In in a mouse model of liver metastases, Frentzas et al [52] showed that by silencing the expression of actin-related protein 2/3 (ARP2/3), a protein complex involved in actin-mediated cell motility, vessel co-option can be inhibited. Interestingly, preclinical trials showed improved tumour control when VEGF and vessel co-option are inhibited simultaneously compared to blocking of VEGF signalling alone[106]. Another novel approach that has been tested in mouse model of brain-metastatic breast cancer and showed some promising results, is inhibition of adhesion of cancer cells to pre-existing blood vessels to block vessel co-option through inhibition of L1 Cell Adhesion Molecule(L1CAM) or the cell adhesion receptor β1 integrin [107].Moreover, pre-clinical models of glioma, and metastases to the liver, lymph nodes or lungs that are vessel co-option dependent, showed that blocking of both angiopoietin and VEGF pathway was more effective compared to inhibition of VEGF alone [108,109]. The exact role of angiopoietin in recruitment or maintenance of co-opted tumour vessels is not fully understood [110]. However, a phase 2 clinical trial of angiopoietin inhibition with and without bevacizumab in recurrent glioblastoma did not show improvement in progression free survival (PFS) [110].

Figure 6 is not clear. It should be   better explained

AR: We have made additions to figure 6 in response to reviewers suggestions     as part of the figure and its legend and within the body of the revised manuscript

Round 2

Reviewer 1 Report

The authors have addressed my suggestions and corrections.

Reviewer 3 Report

The authors have done a very good editing